# Hematological Complications in a COVID-19 Patient: A Case Report

**DOI:** 10.3390/diseases12010005

**Published:** 2023-12-24

**Authors:** Eleonora Ianuà, Mario Caldarelli, Giuseppe De Matteis, Rossella Cianci, Giovanni Gambassi

**Affiliations:** Department of Translational Medicine and Surgery, Catholic University, Fondazione Policlinico Universitario “Agostino Gemelli”, IRCCS, 00168 Rome, Italy; lolaianua@gmail.com (E.I.); mario.caldarelli01@icatt.it (M.C.); giuseppe.dematteis@policlinicogemelli.it (G.D.M.); giovanni.gambassi@unicatt.it (G.G.)

**Keywords:** acquired hemophilia A, SARS-CoV-2 infection, antibodies production, immune dysregulation

## Abstract

Hemophilia A is a hemorrhagic disorder caused by insufficient or inadequate coagulation factor VIII activity. Two different forms are described: congenital, hereditary X-linked, and acquired. Acquired hemophilia A (AHA) is a rare condition and it is defined by the production of autoantibodies neutralizing factor VIII, known as inhibitors. We report the case of a 72-year-old man with a clinical diagnosis of AHA after SARS-CoV-2 infection, which has been described in association with several hematological complications. SARS-CoV-2 infection could represent the immunological trigger for the development of autoantibodies. In our patient, SARS-CoV-2 infection preceded the hemorrhagic complications by 15 days. This lag time is in line with the other cases reported and compatible with the development of an intense immune response with autoantibody production. It is possible that since our patient was affected by type 1 diabetes mellitus, he was more prone to an immune system pathological response against self-antigens. A prompt, appropriate therapeutic intervention with activated recombinant factor VII administration and cyclophosphamide has led to rapid remission of clinical and laboratory findings.

## 1. Introduction

We illustrate the case of a 72-year-old man who presented several spontaneous hematomas 15 days following SARS-CoV-2 infection.

The patient was admitted to our Division of General Internal Medicine at Fondazione Policlinico Universitario “Agostino Gemelli”, IRCCS, Rome and provided written consent for publishing his anonymized data.

## 2. Case Report

The patient, on dual antiplatelet therapy (cardioaspirin 100 mg/die and clopidogrel 75 mg/die) for chronic coronary artery disease for over two years, was admitted to the emergency room of our academic medical center complaining about the appearance of multiple, subsequent, spontaneous hematomas in both legs, arms, and face (Figure 1) over the course of one month. 

The initial hematoma involved the right thigh and was associated with local tenderness and pain. Subsequently, additional hematomas affected other areas, such as arms and the left thigh, without any evidence of even the mildest trauma. 

Treatment with antiplatelet agents had been well tolerated for over two years with no hemorrhagic complications. After consultation with his cardiologist, clopidogrel was discontinued but there was no apparent benefit. For this reason, low-dose aspirin was also discontinued, and the patient was referred for a hematological evaluation.

At the hematology clinic, initial laboratory tests documented: hemoglobin (Hb) 7.9 g/dL, red blood cells (RBC) 2.63 × 10^12^/L, white blood cells (WBC) 8.6 × 10^9^/L, platelets (PLT) 262 × 10^9^/L. Coagulation parameters indicated activated partial thromboplastin time (aPTT) prolongation, factor VIII 1.80% (range 70.0–140.0%). The patient was referred to the acute care hospital to receive a blood transfusion.

Past medical history was relevant for type 1 diabetes mellitus on insulin treatment since the age of 27 years, high blood pressure, hypercholesterolemia, and chronic kidney disease stage III (according to kidney Disease Improving Global Outcomes (KDIGO) classification). The patient had been subjected to surgery for appendicectomy (53 years earlier), meniscectomy, and hip arthroprosthesis (respectively 6 years and 2 years earlier) without experiencing any hemorrhagic complications. He denied any drug or food allergy.

The patient had been vaccinated for SARS-CoV-2 three times but developed a minimally symptomatic infection, diagnosed by SARS-CoV-2 RT-PCR test on nasopharyngeal swab a few months after the last vaccine shot.

Medications included bisoprolol 2.5 mg, losartan 100 mg, atorvastatin 20 mg, ezetimibe 10 mg, omega-3 free fatty acid, clopidogrel 75 mg, cardioaspirin 100 mg, allopurinol 300 mg, pantoprazole 20 mg, and recombinant insulin 15 + 25 + 25 UI along with glargine insulin 28 UI. These medications have been unchanged for the prior 10 years until the discontinuation of the antiplatelet agents.

Upon arrival at the emergency room, blood parameters were the following: Hb 8.2 g/dL, RBC 2.73 × 10^12^/L, mean corpuscular volume (MCV) 88.1 fL, PLTs 325 × 10^9^/L, WBC 10.62 × 10^9^/L. Coagulation tests documented: prothrombin time (PT) 11.1 s, international normalized ratio (INR) 1.01, aPTT 83.2 s, antithrombin III 86%, factor VIII inhibitor 3.8 U/mL, factor VIII 3.2%, detectable inhibitors (range 70.0–140.0%). Two units of blood were transfused and immunosuppressive treatment with methylprednisolone 80 mg intravenous and cyclophosphamide 50 mg was initiated. In order to prevent further bleeding, porcine factor VIII was given at the dosage of 100 UI/kg. At a 12 h interval, aPTT was still prolonged (78 s) and factor VIII was 4.5%. Then, activated recombinant factor VII was initiated at a dosage of 7 mg every 6 h. The patient was then transferred to the general internal medicine division. 

An ultrasound examination of the right thigh documented a single hematoma involving the muscles in the posterior compartment. A total body computed tomography scan performed with spiral and sequential multilayer techniques, before and after i.v. administration. of organo-iodinated contrast medium (Ultravist 370 mg/mL, 120 mL@3 mL/s), showed no sign of organomegaly, masses, or lymphadenopathy, and a fecal occult blood test was repeatedly negative. 

As per clinical routine, serum protein electrophoresis, serum immunoglobulins level, immunoglobulin free light chains, B-2-microglobulin, lymphocyte subpopulation immunophenotyping, serum and urine immunofixation were performed and indicated a minimal monoclonal immunoglobulin G kappa component, deemed worthy only of monitoring through time.

Furthermore, the CD4/CD8 T cell ratio was 1.78 (normal range 1.20–2.40) and the CD19 B lymphocyte count was 519 × 10^9^/L (normal range 100–410).

Blood test count documented a neutrophil count equal to 5960/L (normal range 2.00–7.00) and a monocyte count of 0.6 × 10^9^/L (normal range 0.20–1.00).

A complete serological evaluation included human immunodeficiency virus (HIV), hepatitis C virus (HCV), hepatitis B virus (HBV), cytomegalovirus (CMV), Epstein–Barr virus (EBV), herpes simplex virus (HSV), varicella zoster virus (VZV), beta-d-glucan assay, and quantiferon. HBV resulted positive for hepatitis B anticore antibodies (HbcAb > 8 S/CO), with hepatitis B antigen S antibodies (HbsAb) positive (HbsAb 9.99 mUI/mL). Prophylactic antiviral therapy with lamivudine was initiated (Table 1). Blood lymphocyte levels were monitored daily and when they decreased to a value below 1000/mm^3^, antiviral therapy with acyclovir was added along with trimethoprim/sulfamethoxazole. 

During hospitalization, aPTT and factor VIII activity were monitored but no improvement was observed, and new, although small, subcutaneous hematomas appeared on the left thigh, low back, and shoulder, with no significant change in hemoglobin levels (day 8). Therefore, activated recombinant factor VII administration (7 mg daily) was resumed and cyclophosphamide dosage was increased to 100 mg. At day 13, a slight aPTT and factor VIII activity improvement (aPTT 65.1 s, factor VIII activity 8.1%) was documented and the hemoglobin level was raised to 12.5 g/dL while the patient experienced no new visible hematomas. For such reason, treatment with recombinant activated factor VII was discontinued and after close monitoring for an additional three days, the patient was discharged home with an indication to return to the hematology clinic for follow-up visits (day 17). 

Two weeks after discharge, the patient reported no new onset hematomas. Coagulation parameters were the following: aPTT 39.6 s, factor VIII activity 22.6%, factor VIII inhibitors 6.6 U/mL. Immunosuppressive therapy was progressively tapered down and cardioaspirin was reintroduced, without any hemorrhagic complications.

## 3. Discussion

Hemophilia A is the second most common hemorrhagic disorder caused by insufficient or inadequate coagulation factor VIII activity. Two different forms are described: congenital, hereditary X-linked, and acquired. The latter is a rare condition, and it is defined by the production of autoantibodies neutralizing factor VIII, known as inhibitors.

Although it has been described in people of any age and gender, most cases of acquired hemophilia are either associated with pregnancy or diagnosed in older individuals (>60 years old). About half of the cases are idiopathic, while the remaining cases develop in patients with autoimmune diseases, infections, malignancy, or as a result of drug use [1]. 

AHA is due to an antibody response, usually polyclonal, leading to the production of neutralizing autoantibodies against coagulation factor VIII, which interferes with its functioning [2].

Acquired hemophilia should be clinically suspected in patients who develop new onset hemorrhages, mostly in the form of subcutaneous and/or deep muscular hematomas, without any trauma or with very minimal events. At odds with the congenital form of hemophilia, hemarthrosis is rare [3,4].

Typically, patients show isolated prolonged aPTT, reduced factor VIII activity and circulating neutralizing autoantibodies, detected by Bethesda assay or enzyme-linked immunosorbent assay [5].

The treatment goals are: to stop bleeding, to suppress inhibitor production and to eliminate the clone responsible for its generation [6]. The administration of recombinant active factor VII activated prothrombin complex concentrate, or recombinant porcine factor VIII serve the first goal. Instead, the second requires immunosuppressive therapy with corticosteroids, cyclophosphamide, and/or rituximab until complete remission [5].

It is imperative to prioritize both the rapidity and appropriateness of treatment and management. In case of hemorrhagic events, the primary goal of therapy is to promptly control acute bleeding and eliminate the inhibitor, thereby reducing the risk of recurrent bleeding. This risk persists as long as the anti-FVIII inhibitor is present. Although the treatment of acute bleeding is crucial for AHA, it is important to note that not all patients experience bleeding, and not all cases of bleeding necessitate intervention. Indeed, it is important to prevent the recurrence of bleeding episodes by focusing on the underlying inhibitory factors [7].

Individuals experiencing severe bleeding accompanied by a decline in hemoglobin levels necessitate rapid hemostatic treatment. Conversely, patients with mild to moderate bleeding, with hemoglobin levels not significantly reduced, may not urgently require hemostatic therapy. However, they could potentially progress to severe bleeding at any moment. Patients identified as high-risk for bleeding due to factors such as recent surgery or delivery should undergo prophylactic hemostatic therapy to mitigate the risk of bleeding events [8].

The management of acute bleeding has been significantly improved by the use of bypassing agents, such as activated prothrombin complex concentrates (e.g., factor VIII inhibitor bypassing activity (FEIBA), at a dose of 50–100 U/kg every 8–12 h, up to a maximum of 200 U/kg/24 h). Recombinant activated factor VII (rFVIIa, NovoSeven) represents an alternative at a dose of 90–120 µg/kg every 2–3 h. Another choice can be FVIII porcine at an initial dose of 200 IU/kg, contributing to the remarkable improvement of acute bleeding management [9].

Recently, some studies have demonstrated the effectiveness of Emicizumab in AHA [10]. Emicizumab is an approved therapeutic antibody with a humanized bispecific structure that mimics factor VIII (FVIII). It is used for the prophylaxis of bleeding in individuals with hemophilia, both those with inhibitors and those without [11]. Emicizumab offers advantages due to its subcutaneous administration, prolonged hemostatic effectiveness, and anticipated low thrombogenicity [12].

Nevertheless, evaluating coagulation function while under the influence of Emicizumab is difficult, and its current approval does not extend to AHA [10].

Another crucial consideration is the decision to initiate treatment, considering the fact that the proinflammatory condition triggered by the infection increases the likelihood of thrombosis. Addressing bleeding in AHA requires the application of bypassing agents. Nevertheless, it is essential to highlight that the use of these agents is associated with an increased risk of prothrombotic events, especially in individuals with pre-existing risk factors or recent thromboembolic occurrences [5].

In individuals with severe bleeding disorders by AHA, addressing both bleeding and thrombosis presents significant challenges. The treatment decision often relies on evaluating the specific hemorrhagic and thromboembolic risks of each patient. Given the limited data available for AHA, insights can be gleaned from patients with congenital hemophilia, both with and without inhibitors [13].

For individuals with congenital hemophilia, the consensus is that the use of antiplatelets and anticoagulants is typically contraindicated. This restriction narrows the therapeutic options when facing scenarios such as thromboembolism or the need for anticoagulation in these patients.

Acquired hemorrhagic manifestations may be associated with many different causes, and in patients receiving multiple medications, they can be an adverse effect of drugs, like antiplatelet agents and/or anticoagulants. At the onset of the symptoms, our patient was on dual antiplatelet therapy for coronary artery disease and those medications were initially considered responsible for the bleeding despite having been well tolerated for many years before. Indeed, due to its rarity, the diagnosis of AHA is often overlooked. Furthermore, in the cases described previously, AHA presented between 3 weeks and 3 months after initiation of the treatment [14]. Since our patient had been taking clopidogrel for nearly ten years without any complication, a drug-induced AHA does not seem to be a plausible hypothesis. 

The reason why our patient developed AHA is not clear. In the last few years, cases of AHA in COVID patients have been described, both after infection and vaccination [15,16,17,18,19,20,21,22,23,24,25]. A few cases of AHA following the viral infection have been reported in literature [15,26,27,28,29,30]. Our first hypothesis is that SARS-CoV-2 infection acted as the immunological trigger for the development of autoantibodies. SARS-CoV-2 infection has been associated with several hematological complications, including autoimmune hemolytic anemia, and lupus anticoagulants.

A retrospective study in hemophilia A patients reported that COVID-19 is linked to a higher risk of bleeding and hospitalizations [31] due to the possible development of FVIII inhibitors after an immune trigger [20].

The mechanism through which SARS-CoV2 might have activated an immunologic response suggests molecular mimicry phenomena between the virus and human proteins: the infection activates an immune response against the virus during the convalescence phase that may cross-react with the host’s proteins due to peptide sequencing sharing [32]. 

Cases of AHA after SARS-CoV2 vaccination have also been described, but the exact mechanism of correlation is controversial. The rate of acquired hemophilia post-vaccination is not different from that without vaccination [33].

Specific sequences within the SARS-CoV2 spike protein, which coincide with highly immunogenic FVIII sequences, have been pinpointed. These sequences constitute part of the vaccine’s antigenic composition and may potentially interact with T cell clones that have not been eliminated and are specific to the FVIII. These autoreactive T cell clones have been detected in individuals susceptible to developing AHA. The stimulation of these T cell clones can, in turn, trigger the maturation and activation of B cell clones, leading to the production of natural anti-FVIII antibodies. Notably, this mechanism remains plausible even when considering the temporal relationship between immunization and the diagnosis of AHA [34].

Our patient had been infected with SARS-CoV-2 virus in the first half of December, while symptoms started to appear in January. The time interval from the infection to the development of the first signs and symptoms was in line with the other cases described in literature and with the development of an intense immune response with autoantibody production. Moreover, our patients suffered from type 1 diabetes mellitus, an autoimmune disease that suggests an immune system predisposition to pathological responses against self-antigens.

It has also been proposed that vaccination could induce an autoimmune reaction through antigenic mimicry and the activation of dormant auto-reactive T and B cells, resulting in the formation of autoantibodies [35].

While establishing a definitive causal relationship between vaccines and AHA poses considerable challenges, and the occurrence of FVIII inhibitors post-vaccination or infection may be coincidental, we believe that there is a reasonable association in the reported case. This association is based on a comprehensive analysis of anamnestic, clinical, and biological data, with the consideration of the timing between symptom onset and vaccination. 

Probably, the predisposition to autoimmune pathologies in clinical history provided the substrate for the onset of AHA, following exposure to vaccination and subsequently to the SARS-CoV-2 virus.

## 4. Conclusions

AHA is an immune-mediated hematological disorder, characterized by the presence of neutralizing autoantibodies against coagulation factor VIII activity. Several factors, such as neoplasms, autoimmune and infectious diseases, and pregnancy, have been linked to its onset [36]. A few cases of AHA following SARS-CoV-2 infection have been reported in the literature [36], probably linked to the cytokine storm and the systemic immune dysregulation, during acute infection. In our patient, SARS-CoV-2 infection preceded the hemorrhagic complications by 15 days. This lag time is in line with the other cases reported and compatible with the development of an intense immune response with autoantibodies production. It is possible that since our patient was affected by type 1 diabetes mellitus, he was more prone to an immune system pathological response against self-antigens. A prompt, appropriate therapeutic intervention with activated recombinant factor VII administration and cyclophosphamide led to the rapid remission of clinical and laboratory findings.

At present, there is insufficient clinical evidence to establish a cause–effect relationship between SARS-CoV-2 infection and AHA. The potential link between SARS-CoV-2 infection and AHA is intriguing, albeit not unexpected, given the documented immune dysregulation following SARS-CoV-2 infection. Another pathophysiological mechanism of SARS-CoV-2 infection-induced autoimmunity involves the activation of dormant autoreactive T and B cells, alongside molecular mimicry.

## Figures and Tables

**Figure 1 diseases-12-00005-f001:**
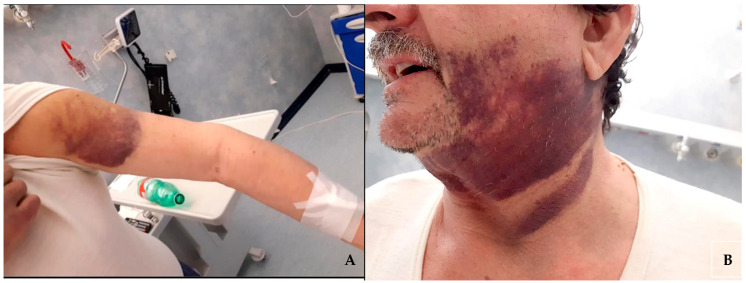
Hematomas on the patient’s left arm (**A**) and of the face and neck region (**B**).

**Table 1 diseases-12-00005-t001:** The table summarizes the main results of the laboratory and instrumental tests performed.

Laboratory Tests	Value	Normal Range
HbsAb	9.99 mUI/mL	>10 mUI/mL
HbcAb	>8.00 S/CO	>1.00 S/CO positive
Hepatitis C Virus	0.02 S/CO	>1.10 S/CO positive
Cytomegalovirus	Not detectable	
Epstein-Barr Virus	<40 UI/mL	>40 = significant
Herpes Simplex Virus IgG	<0.5 UI/L	>1.10 UI/L positive
Varicella Zoster Virus IgG	1366.5 UI/L	>110 UI/L Positive
Human Immunodeficiency Virus	Not detectable	
CD4/CD8 T cell ratio	1.78	1.20–2.40
CD19 B lymphocyte	532 × 10^9^/L	100–410 × 10^9^/L
Neutrophils count	6.58 × 10^9^/L	2.00–7.00 × 10^9^/L
Monocytes count	0.56 × 10^9^/L	0.20–1.00 × 10^9^/L
**Instrumental Tests**	**Findings**	
Ultrasound examination of the right thigh	Single hematoma involving the muscles of the posterior compartment	
Total body Computed Tomography scan	No sign of organomegaly, masses, or lymphadenopathy	

## Data Availability

The data presented in this study are available on request from the corresponding author. The data are not publicly available due to privacy.

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
