# Peer review of "Hematological Complications in a COVID-19 Patient: A Case Report"

_diseases, 2023, doi:10.3390/diseases12010005_

Round 1
Reviewer 1 Report
Comments and Suggestions for Authors
The title has an unusual term "Immunoemathological" . A suggestion is that it is intended to be "hematological" complications.
Another noted a spell check issue "right tight" in lines 37, 75 and "Left tight" in line 93. This must be "thigh"?
Author Response
Rome, 18th December 2023
Dear Editor of “Diseases”,
first of all, my coauthors and I would like to thank You sincerely for this opportunity of cooperation, following the submission of the paper “Immunoemathological complications in a covid-19 patient: a case report” and its possible publication upon “Diseases”.
We profoundly thank the reviewers for the comments and useful suggestions aimed at improving the final version of the paper.
This is a point-by-point list of changes made in the paper:
REVIEWER 1
The title has an unusual term "Immunoemathological". A suggestion is that it is intended to be "hematological" complications.
Thank you for your suggestion. We have changed the title as suggested.
Another noted a spell check issue "right tight" in lines 37, 75 and "Left tight" in line 93. This must be "thigh"?
Thank you for your suggestion. We corrected errors throughout the text.
We thank You for your constructive critique and we hope the review process has led to an improved manuscript.
If additional changes are warranted, we will make them.
We hope that this revised version of our manuscript may now be found suitable for publication.
Sincerely,
Rossella Cianci, MD, PhD
Reviewer 2 Report
Comments and Suggestions for Authors
The case report presents possible connection between a patient who is type 1 diabetic, and showed Covid19 symptoms who developed symptoms of AHA, although report is interesting but there are several observation from this reviewer that needs clarification.
Major
1. It is mentioned that patient showed mild symptoms of covid19 but whether diagnosis was confirmed ( rt-pcr Or any method) is not refereed in the manuscript.
2. There is no mention of patient consent or ethical clearance obtained for reporting this case is mentioned.
3. Some of the tests performed mentioned in line 79-82 , no reference, no description of method and instruments used described in the MS.
4. Results of the tests described in 79-82 should be presented in a tabulae format, otherwise what the results indicated is not clear.
5. It not at all clear wheather AHA developed is linked to supposed sars-Cov infection is linked or not. As patient already have autoimmune type 1 diabetes. Also immunological findings like CD4/CD8 T cell ratio, Neutrophil count, monocyte count, B cell count is not described so hard to discern specific activation of autoimmune response.
6. Biggest concern is there is no clear evidence to justify the title, In fact title has misspelling, it should be immunohematological .
Minor:
In abstract line 11 and 12 is repeated.
In fact there seems to be a lack of care while preparing the manuscript. Should be thoroughly edited.
Comments on the Quality of English LanguageTypological error and misspelling is common in this manuscript, should be edited thoroughly.
Author Response
Rome, 18th December 2023
Dear Editor of “Diseases”,
first of all, my coauthors and I would like to thank You sincerely for this opportunity of cooperation, following the submission of the paper “Immunoemathological complications in a covid-19 patient: a case report” and its possible publication upon “Diseases”.
We profoundly thank the reviewers for the comments and useful suggestions aimed at improving the final version of the paper.
This is a point-by-point list of changes made in the paper:
REVIEWER 2
The case report presents possible connection between a patient who is type 1 diabetic, and showed Covid19 symptoms who developed sy<mptoms of AHA, although report is interesting but there are several observations from this reviewer that needs clarification.
Major
- It is mentioned that patient showed mild symptoms of covid19 but whether diagnosis was confirmed (rt-pcr Or any method) is not refereed in the manuscript.
Thank you for your suggestion. We have added the method used for diagnosis.
- There is no mention of patient consent or ethical clearance obtained for reporting this case is mentioned.
As you requested, we have added a sentence about patient consent at the end of introduction. Thank you
- Some of the tests performed mentioned in line 79-82, no reference, no description of method and instruments used described in the MS.
We have included the description of some instrumental tests performed.
- Results of the tests described in 79-82 should be presented in a tabulae format, otherwise what the results indicated is not clear.
We have inserted a table that presents the results of the tests mentioned in the lines as indicated.
- It not at all clear whether AHA developed is linked to supposed sars-Cov infection is linked or not. As patient already have autoimmune type 1 diabetes. Also immunological findings like CD4/CD8 T cell ratio, Neutrophil count, monocyte count, B cell count is not described so hard to discern specific activation of autoimmune response.
We have added the requested data and discussed about the possible link between AHA and SARS-CoV-2 infection.
- Biggest concern is there is no clear evidence to justify the title, In fact title has misspelling, it should be immunohematological.
We corrected the title.
Minor:
In abstract line 11 and 12 is repeated.
We deleted line 11.
In fact there seems to be a lack of care while preparing the manuscript. Should be thoroughly edited.
Typological error and misspelling is common in this manuscript, should be edited thoroughly.
We have corrected typos throughout the manuscript.
We thank You for your constructive critique and we hope the review process has led to an improved manuscript.
If additional changes are warranted, we will make them.
We hope that this revised version of our manuscript may now be found suitable for publication.
Sincerely,
Rossella Cianci, MD, PhD
Reviewer 3 Report
Comments and Suggestions for Authors
Please correct the typographical and spelling errors.
The paper is interesting in linking AHA in a vaccinated patient with COVID.
Given the link between autoimmunity and COVID vaccination (particularly the adenoviral vaccines), the paper may be valuable, but whether this was a spontaneous case, or was linked to the vaccine, or the disease, is not clear. Moreover, there have been many cases reported of AHA following COVID or vaccination. As such, the current paper is not novel, but does confirm observations of others.
Among the millions of recipients of the COVID vaccines, as well as the millions who have developed COVID, there have undoubtedly been a significant number of individuals with underlying autoimmune diseases such as type I diabetes, MS, lupus, etc. It would be interesting for the authors to further speculate why this particular patient developed that auto antibodies.
The authors should proof read the paper and correct the grammatical and spelling errors that are found throughout the paper.
Comments on the Quality of English Language
The paper contained several typos and should be edited.
Author Response
Rome, 18th December 2023
Dear Editor of “Diseases”,
first of all, my coauthors and I would like to thank You sincerely for this opportunity of cooperation, following the submission of the paper “Immunoemathological complications in a covid-19 patient: a case report” and its possible publication upon “Diseases”.
We profoundly thank the reviewers for the comments and useful suggestions aimed at improving the final version of the paper.
This is a point-by-point list of changes made in the paper:
REVIEWER 3
Please correct the typographical and spelling errors.
We have corrected errors throughout the paper
The paper is interesting in linking AHA in a vaccinated patient with COVID.
Given the link between autoimmunity and COVID vaccination (particularly the adenoviral vaccines), the paper may be valuable, but whether this was a spontaneous case, or was linked to the vaccine, or the disease, is not clear. Moreover, there have been many cases reported of AHA following COVID or vaccination. As such, the current paper is not novel, but does confirm observations of others.
Among the millions of recipients of the COVID vaccines, as well as the millions who have developed COVID, there have undoubtedly been a significant number of individuals with underlying autoimmune diseases such as type I diabetes, MS, lupus, etc. It would be interesting for the authors to further speculate why this particular patient developed that autoantibodies.
Thank you for your suggestion.
We have added considerations about the possible relationship between the two pathologies in this patient.
The authors should proof read the paper and correct the grammatical and spelling errors that are found throughout the paper.
The paper contained several typos and should be edited.
We have corrected errors throughout the paper
We thank You for your constructive critique and we hope the review process has led to an improved manuscript.
If additional changes are warranted, we will make them.
We hope that this revised version of our manuscript may now be found suitable for publication.
Sincerely,
Rossella Cianci, MD, PhD
Round 2
Reviewer 2 Report
Comments and Suggestions for Authors
Inclusion of information and editing improved quality of MS significantly